# Escaping KRAS: Gaining Autonomy and Resistance to KRAS Inhibition in KRAS Mutant Cancers

**DOI:** 10.3390/cancers13205081

**Published:** 2021-10-11

**Authors:** Yuta Adachi, Ryo Kimura, Kentaro Hirade, Hiromichi Ebi

**Affiliations:** 1Division of Molecular Therapeutics, Aichi Cancer Center Research Institute, Nagoya 464-8681, Japan; y.adachi@aichi-cc.jp (Y.A.); r.kimura@aichi-cc.jp (R.K.); k.hirade@aichi-cc.jp (K.H.); 2Division of Advanced Cancer Therapeutics, Graduate School of Medicine, Nagoya University, Nagoya 466-8650, Japan

**Keywords:** KRAS, dependency, autonomy, EMT, metabolic reprogramming, YAP1, RSK

## Abstract

**Simple Summary:**

While KRAS is a driver oncogene, tumor cells can acquire mutant KRAS independency by activating pathways that functionally substitute for mutant KRAS. These KRAS-independent tumor cells exhibit a mesenchymal phenotype, readily primed for potential metastasis. The activation of YAP and/or RSK-mTOR pathways and mutations in LKB1, KEAP1, and/or NRF2 are associated with mutant KRAS autonomy. These alterations rewire survival signaling and metabolic processes originally governed by mutant KRAS. The presence of KRAS-independent cells is associated with the heterogeneity of KRAS mutant cancers, as well as variable responses to therapies. Notably, KRAS G12C-specific inhibitors appear to be effective only in tumors dependent on mutant KRAS for their survival. Therefore, determining KRAS dependency will be critical for selecting patients who should be treated with mutant-specific inhibitors. Furthermore, elucidating underlying mechanisms of KRAS autonomy is crucial towards developing optimal treatment strategies for KRAS-independent tumors.

**Abstract:**

Activating mutations in KRAS are present in 25% of human cancers. When mutated, the KRAS protein becomes constitutively active, stimulating various effector pathways and leading to the deregulation of key cellular processes, including the suppression of apoptosis and enhancement of proliferation. Furthermore, mutant KRAS also promotes metabolic deregulation and alterations in the tumor microenvironment. However, some KRAS mutant cancer cells become independent of KRAS for their survival by activating diverse bypass networks that maintain essential survival signaling originally governed by mutant KRAS. The proposed inducers of KRAS independency are the activation of YAP1 and/or RSK-mTOR pathways and co-mutations in *SKT11* (LKB1), *KEAP1*, and *NFE2L2* (NRF2) genes. Metabolic reprogramming, such as increased glutaminolysis, is also associated with KRAS autonomy. The presence or absence of KRAS dependency is related to the heterogeneity of KRAS mutant cancers. Epithelial-to-mesenchymal transition (EMT) in tumor cells is also a characteristic phenotype of KRAS independency. Translationally, this loss of dependence is a cause of primary and acquired resistance to mutant KRAS-specific inhibitors. While KRAS-dependent tumors can be treated with mutant KRAS inhibitor monotherapy, for KRAS-independent tumors, we need an improved understanding of activated bypass signaling pathways towards leveraging vulnerabilities, and advancing therapeutic options for this patient subset.

## 1. Introduction

KRAS is one of the most frequently mutated oncogenes in human cancer, occurring in 95% of pancreatic, 50% of colorectal, and 32% of lung adenocarcinomas, with descending prevalence in many other tumor types [1]. The KRAS small GTPase cycles between an active guanosine triphosphate (GTP)-bound and an inactive guanosine diphosphate (GDP)-bound state that functions as an ON-OFF molecular switch. RAS cycling is mediated by guanine nucleotide exchange factors (GEFs, e.g., SOS1/2) that exchange GDP for GTP, and guanine nucleotide activating proteins (GAPs, e.g., NF1) that catalyze the hydrolysis of GTP to GDP. Activated RAS-GTP binds preferentially to downstream effectors containing a RAS-binding-domain (RBD) or RAS-association (RA)-domain. There are at least 11 unique RAS effector families, each of which activates a distinct protein signaling cascade. Four families—mitogen-activated protein kinase (MAPK), phosphatidylinositol 3-kinase (PI3K), Ral, and TIAM1—have driving roles in oncogenesis [2,3].

While RAS is predominantly in its GDP-bound inactive state in normal quiescent cells, mutations in KRAS render it persistently GTP-bound and constitutively active, independent of extracellular stimuli, resulting in the activation of effector signaling pathways that drive cancer growth [3,4]. Adenosine triphosphate (ATP)-competitive inhibitors targeting other driver oncogenes, such as *EGFR* and *ALK*, have been successful. However, due to the picomolar affinity of GTP for KRAS, and the millimolar concentrations of cellular GTPs, the development of direct GTP inhibitors has been challenging [5,6]. Therefore, indirect strategies for the inhibition of downstream effector signaling pathways have been evaluated. Unfortunately, these methodologies have demonstrated suboptimal efficacies. Furthermore, while recent success in developing KRAS G12C inhibitors has proved that mutant KRAS is, in fact, druggable, the clinical activity of KRAS G12C inhibitors is underwhelming when compared to EGFR and ALK inhibitors [7]. Unlike patients with lung cancer harboring EGFR mutations or ALK translocations, diseases for which targeted agents typically achieve 60 to 80% response rates, KRAS G12C inhibitors such as sotorasib and adagrasib typically yield 40 to 50% response rates [8]. One possible reason for the imperfect response comes from the heterogeneity among KRAS mutant cancers. In this review, we first describe the heterogeneity that is at the center of the dependency to mutant KRAS and the epithelial-to-mesenchymal transition (EMT) of these tumor cells. We then discuss how EMT plays a role in the discordant response to drugs targeting KRAS mutant cancers, and the possible ways to overcome it.

## 2. Phenotypes Generated by Different Oncogenic Mutations in KRAS

The vast majority of KRAS alterations are single nucleotide point mutations, most of which are identified at codons 12, 13, and 61. Among them, codon 12 is by far the most frequently mutated. There are substantial differences in the biological and clinical behavior of each common variant. For example, while mutations at codons 12, 13, and 61 all reduce the rate of intrinsic and GAP-mediated hydrolysis, an increased rate of intrinsic and GEF-mediated nucleotide exchange was observed in G13 and Q61 mutants, but not in G12 mutants [9]. In addition, the Q61L and Q61H RAS mutants exhibit the lowest intrinsic GTP hydrolysis rates [10]. Besides these hot spots, A146 mutations increase the rate of GDP exchange without affecting the rate of GTP hydrolysis [11]. These differences in the production of GTP-bound KRAS precipitate the distinct activation of effector signaling. Among KRAS mutant lung cancer cell lines, KRAS G12D preferentially activates MAPK signaling, while KRAS G12C or G12V mutants activate Ral signaling and exhibit decreased growth factor-dependent activation of the PI3K pathway [12].

The diverse activation of effector signaling by each KRAS variant may affect the sensitivity to targeted therapies. For example, anti-EGFR therapy is excluded in patients with KRAS mutant colorectal cancer due to receptor tyrosine kinase (RTK)-independent activation of mutant KRAS. However, some reports suggest that KRAS G13D mutant colorectal cancer is associated with an improved outcome after treatment with the anti-EGFR antibody cetuximab compared to responses observed for other KRAS mutations [13]. Mechanistically, NF1 GAP proteins stimulate GTP hydrolysis when bound to KRAS G13D, while the protein is inactive against G12- and Q61-mutated KRAS. Since NF1 is frequently co-mutated in KRAS G13-mutated colorectal cancer, these tumor cells can respond to EGFR inhibitors in an NF1-dependent manner [14]. While these reports indicate that different KRAS mutant alleles can contribute to the heterogeneity of KRAS mutant cancers, specific KRAS alleles have not been consistently associated with a subtype of KRAS mutant lung cancer [15]. Therefore, the relationship between each mutant KRAS allele and the magnitude of the role of mutant KRAS in tumor cell survival are not fully characterized.

## 3. KRAS Dependency

Oncogene addiction is a biological phenomenon where the survival and proliferation of cancer cells depend on the continued expression of the oncogene. In fact, in cancers harboring driver oncogenes, such as mutant *EGFR* in lung cancer and mutant *BRAF* in melanoma, studies have demonstrated that knockdown or knockout of the oncogene induces robust apoptosis [16,17]. However, in the case of KRAS mutant cancer, the effect of KRAS depletion results in the variable induction of apoptosis, highlighted in KRAS mutant lung and pancreatic cancer cell lines [18]. These reports appear inconsistent with earlier studies that showed considerable apoptosis caused by the disappearance of mutant Kras induced either by temperature changes in temperature-sensitive Ras-transformed cells or by the injection of Ras-neutralizing antibodies [19].

One possible explanation for this discrepancy comes from the models used to evaluate mutant KRAS dependency. The oncogenic role of mutant KRAS has been largely investigated in genetically engineered mouse models. For example, in one model, doxycycline regulatable pancreatic expression of G12D mutant Kras with inactivation of one allele of the tumor suppressor gene *p53* generated pancreatic ductal adenocarcinoma (PDAC). Subsequent inactivation of Kras by halting doxycycline administration in these tumor-bearing mice exhibiting signs of sickness, weight loss, and a deteriorating clinical condition ultimately returned the mice to a good health condition. Furthermore, pancreatic tumors became atrophic with rare activation of MAPK signaling [20]. In contrast, KRAS depletion experiments in human models began to separate tumor cells with KRAS dependency from those lacking this reliance. This KRAS independency was eventually determined to be the result of more complicated gene abnormalities and co-occurring alterations in human cell lines. For example, secondary mutations in tumor suppressors, such as *STK11* and *CDKN2A*, contribute to the KRAS independency in lung cancer cell lines [21]. Hence, the concept of KRAS dependency only began to emerge when investigators moved from studying the more genetically simple engineered mouse models onto the more (epi)genetically complex human models.

As we describe below, KRAS-independent tumors can also emerge in Kras mutant mouse models, discovered by long-term observation following mutant Kras depletion. These tumors acquired KRAS independency via various mechanisms. Therefore, it is plausible that KRAS is the major driver gene essential for carcinogenesis, but that it is not necessarily required for tumor maintenance in cases where essential survival signaling is maintained by other means. Notably, tumor cells show negligible apoptosis by KRAS depletion in 2D culture, but become highly dependent on KRAS in 3D culture or in vivo [22]. Therefore, KRAS independency may be overestimated given that the majority of KRAS dependency experiments have been evaluated in 2D culture.

## 4. Escaping Mechanisms of KRAS Dependency

### 4.1. KRAS Rescue Screens

Efforts have been made to identify how tumors acquire their independence from mutant KRAS. A genome-wide genetic rescue screen led to the identification of YAP1 as a survival mechanism in the HCT-116 KRAS dependent colon cancer cell line upon suppression of KRAS [23]. In a doxycycline-inducible Kras G12D mutant pancreatic mouse model, long-term follow-up for 9 to 47 weeks after doxycycline withdrawal resulted in tumor relapse in 70% of mice, with morphological changes from well-differentiated ductal features to poorly differentiated or sarcomatoid features. Array CGH of relapsed tumors identified *YAP1* amplification, and enforced expression of YAP1 enabled tumor maintenance following doxycycline withdrawal [24].

Another genome-wide genetic rescue screen revealed that autocrine IGF1/AKT signaling was a common survival mechanism following KRAS depletion in a doxycycline-inducible Kras G12D pancreatic cancer mouse model [25]. CRISPR knockout of KRAS in pancreatic cell lines also led to PI3K dependence in surviving cells, and this was accompanied by the activation of MAPK signaling through PI3K [26]. Whereas both MAPK and PI3K are well known major effector pathways of KRAS, PI3K was shown to be dominantly regulated by RTKs, chiefly IGFR in colorectal and lung cancer [27,28]. In the case of YAP1-activated cells, MAPK and PI3K were not activated in pancreatic cancer models resistant to Kras depletion, while these pathways were hyperactive in colon and lung cancer models, suggesting lineage-specific rewiring of these pathways.

### 4.2. KRAS Dependency and Its Effector Signaling Pathways

As discussed above, bypass activation of KRAS effector signaling pathways, such as PI3K and MAPK, is a common mechanism of mutant KRAS independency. Besides PI3K and MAPK, KRAS binds and activates a number of other effector proteins, including RalGDS. The involvement and contribution of each effector signaling network to distinct KRAS dependencies have been assessed. In one siRNA screen, combinatorial knockdown was used to evaluate the dependency of effector nodes [21]. For example, the RAF node was assessed by simultaneous knockdown of ARAF, BRAF, and CRAF. The screen resulted in two major classifications of KRAS mutant cell lines: KRAS-dependent, and also dependent on the RAF/MAPK pathway, and KRAS-independent, instead dependent on the p90 ribosomal S6 kinase (RSK)-mTOR pathway. The activation of the RSK-mTOR network appears to be accomplished by PI3K signaling. A third classification involved a small subset of cells that were moderately KRAS-dependent, but strongly dependent on RalGDS and PI3K signaling. Surprisingly, mutant KRAS-independent cells still retain their dependence on the wild-type RAS isoform, suggesting that H/NRAS activate signaling required to maintain cell proliferation and survival in the absence of mutant KRAS.

### 4.3. KRAS Dependency and Metabolism

Mutant KRAS deregulates key metabolic processes, such as glycolysis, glutaminolysis, autophagy, and macropinocytosis [29]. Prominently, oncogenic KRAS alters glucose metabolism to support tumor growth, proliferation, and survival [29]. KRAS promotes glucose uptake and enhances glycolysis by upregulating glucose transporters such as GLUT1 and key glycolysis enzymes, including hexokinase-1/2 and phosphofructokinase 1. KRAS also activates the hexosamine biosynthesis pathway and the non-oxidative arm of the pentose phosphate pathway to promote the increase in glycolysis, and to support tumor cell viability [30]. The activation of the pentose phosphate pathway generates ribose-5-phosphate for de novo nucleotide biosynthesis. Glucose metabolism is mainly regulated by mutant KRAS-mediated MAPK and MYC signaling [30]. Mutant KRAS also promotes glutaminolysis, a process that uses steps from the tricarboxylic cycle (TCA) to convert glutamine to α-ketoglutarate and other molecules. Oncogene-addicted cancers, including KRAS mutant cancers, are known to be dependent on glutamine for their survival [31]. In KRAS mutant lung and pancreatic cancer, mutant KRAS transcriptionally activates NRF2, a principal modulator of cellular redox, resulting in a reduced intracellular ROS level and leading to increased glutamine dependency [32,33].

To gain KRAS independency, KRAS mutant cancer cells rewire signals related to glycolysis and glutaminolysis. YAP1 signaling plays important roles in metabolism regulation, such as the promotion of glycolysis, lipogenesis, and glutaminolysis [34]. Therefore, KRAS independency induced by the activation of YAP1 signaling is related to the rewiring of metabolic processes in KRAS mutant cancers. KRAS mutant lung cancers have been divided into three groups by integrating genomic, transcriptomic, and proteomic analyses: TP53-inactivated, CDKN2A/B-inactivated, and LKB1-inactivated [15]. LKB1-inactivated KRAS mutant lung cancers harbor co-mutations in *KEAP1* and/or significant co-occurrences of copy number loss in *STK11/LKB1* and *KEAP1* due to the chromosomal proximity of these genes [15]. KEAP1 is a negative regulator of NRF2, and aberrant activation of the KEAP1/NRF2 pathway can lead to increased amounts of glutamine produced independently of mutant KRAS [33]. Furthermore, loss of LKB1 can also promote glutaminolysis and the activation of the TCA cycle, partly mediated by NRF2 in KRAS mutant lung cancer [35]. Collectively, co-mutations of LKB1 and/or KEAP1 with KRAS in lung cancer are associated with KRAS independency through a rewiring of metabolic processes (Figure 1). In a mouse model of Kras mutant pancreatic cancer, LKB1 loss instead resulted in mTOR-dependent induction of the serine-glycine-one-carbon (SGOC) pathway coupled to S-adenosylmethionine (SAMe) generation [36]. Therefore, while LKB1 is involved in cancer metabolism, the metabolic pathways regulated by this protein can vary based on tumor lineage.

Mitochondrial metabolism and ROS generation are also essential for Kras-mediated tumorigenicity and the proliferation of tumor cells [37]. In doxycycline-inducible Kras G12D mutant pancreatic mouse models, surviving cells following Kras withdrawal relied on oxidative phosphorylation and cancer stem cell phenotypes for their survival [38]. These cells upregulated key modulators of mitochondrial function, autophagy, and lysosome activity, decreased metabolic intermediates involved in the TCA cycle, and impaired glycolysis. In the siRNA screen for RAS effector signaling networks, the RSK-mTOR pathway-activated tumor cells showed increased activation of mTOR signaling that negatively regulated autophagy, instead elevating oxidative phosphorylation and mitochondrial ribosome maintenance [21]. Notably, low LKB1 activity induced the RSK activating state by de-repressing mTOR and driving oxidative phosphorylation, while high LKB1 activity favored the KRAS-state by promoting glycolysis [21].

To maintain tumor cell viability and survival, oncogenic KRAS induces autophagy, a lysosome-mediated process whereby cells degrade organelles and macromolecules, with the resulting breakdown products subsequently utilized as metabolic intermediates [39]. Furthermore, oncogenic KRAS promotes macropinocytosis to transport extracellular proteins as an amino acid source for the TCA cycle to sustain tumor growth [40]. The autophagy-to-macropinocytosis switch depends on NRF2-mediated induction of macropinocytosis components. Therefore, while the inhibition of the autophagy process can impair the growth of pancreatic cancer cell lines and PDAC development, combined inhibition of autophagy and macropinocytosis has been demonstrated to be a necessary requirement for tumor regression in Kras G12D pancreatic cancer models [40]. These results suggest that the modulation of autophagy and macropinocytosis may also be involved in the process of KRAS independency induced by aberrant activation of the KEAP1/NRF2 pathway.

### 4.4. KRAS Dependency and Epithelial-to-Mesenchymal Transition (EMT)

Importantly, KRAS-dependent cells tend to have an epithelial morphology, whereas KRAS-independent cells typically appear to be more mesenchymal in nature [18,21,41]. In epithelial-like KRAS-dependent cancer cells, the induction of EMT by EMT transcription factors such as ZEB1, Snail, and Twist resulted in resistance to KRAS depletion [42]. Among mechanisms that induce KRAS independency, RSK was shown to be indispensable to the promotion of mesenchymal motility and invasive capacities in non-transformed breast and colon epithelial cells, as well as in carcinoma cells [43]. In addition, YAP/TAZ not only functions downstream of EMT, but also serves as an active inducer of this process [44]. KRAS and YAP1 converge on the transcription factor FOS, which upregulates the transcription of EMT-related genes [23]. Moreover, several metabolites involved in glycolysis and the TCA cycle affect epigenetic reprogramming, leading to the activation of EMT transcription factors [45,46]. As in the case of YAP, EMT can also affect the expression of metabolic genes regulating glucose, glutamine, and nucleotide metabolism [46,47]. These results suggest that EMT is both a cause and a result of KRAS independency, although the precise mechanism of how KRAS mutant cells acquire their mesenchymal phenotype is not yet fully understood.

EMT also plays important roles in the carcinogenesis of KRAS mutant lung cancer. In human bronchial epithelial cells, EMT induced by ZEB1 expression was an early, critical event in the pathogenesis of KRAS mutant lung cancer [48]. Similarly, EMT accelerated Kras G12D lung tumorigenesis by upregulating the expression of key enzymes in the hexosamine biosynthesis pathway. The activation of this pathway elevated levels of O-linked β-N-acetylglucosamine posttranslational modification of intracellular proteins, which suppress oncogene-induced senescence caused by mutant KRAS. Conversely, the suppression of this pathway delayed Kras G12D lung tumorigenesis [49]. Given that the EMT process is reversible, tumor cells may switch back and forth between epithelial and mesenchymal status in early tumorigenesis before transforming into a more permanent mesenchymal state after activating bypass signaling networks that permit full independence from KRAS.

## 5. Treatment Strategies for KRAS Mutant Cancers Based on Mutant KRAS Dependency

### 5.1. KRAS Dependency and Response to Chemotherapy and Molecular Targeting Agents for Effector Signaling Pathways

Translationally, it is important to determine how this heterogeneity should be taken into consideration to develop therapies targeting KRAS mutant cancers. In a mouse model of Kras G12D/Tp53 pancreatic cancer, tissue-specific deletion of the EMT-inducing transcription factors Snail, Twist, or TGF-β receptor 2 (TGFβR2) enhanced the sensitivity of tumors to gemcitabine, suggesting a role for EMT in the resistance to conventional chemotherapy [50]. Notably, the absence of EMT-inducing transcription factors did not affect the number of metastases in these tumors, in contrast to their involvement in the resistance to this chemotherapeutic regimen.

With the exception of KRAS G12C, inhibiting mutant KRAS directly still remains a challenge. Hence, targeting effector signaling continues to be an attractive strategy for treating these cancers. The inhibition of MAPK signaling has been intensively studied among the mutant KRAS effector pathways. However, MEK inhibitor monotherapy has demonstrated only modest efficacy in vitro and in vivo [51,52]. The reason for this insensitivity stems from the reactivation of MAPK signaling following MEK inhibition. MEK-activated ERK phosphorylates hundreds of substrates, including cytoplasmic signaling proteins and nuclear transcriptional factors, thereby regulating a wide array of cellular events. ERK also feedback inhibits the pathway directly by phosphorylating negative regulatory sites on RAF and RTKs, and indirectly by activating the transcription of negative regulators, including the dual-specificity phosphatase (DUSP) proteins, which modulate ERK activity, and the Sprouty (SPRY) proteins, which inhibit RTKs. Thus, the suppression of ERK activity mediated by MEK inhibition relieves this naturally occurring negative feedback, leading to renewed upstream signaling [52,53].

In KRAS mutant lung cancer, EMT has been shown to play a defining role in the feedback reactivation of RTK signaling following MEK inhibition. In epithelial-like cells, this feedback was mediated by Erb-B2 Receptor Tyrosine Kinase 3 (ERBB3). In contrast, mesenchymal-like KRAS mutant lung cancer cells do not express ERBB3. Instead, the feedback was attributed to the fibroblast growth factor receptor 1 (FGFR1) pathway, which is dominantly expressed in mesenchymal-like cells. The suppression of SPRY proteins by MEK inhibition relieved the negative feedback control of basal FGFR-FRS2 function, resulting in the reactivation of MAPK signaling via FGFR1. The combination of the MEK inhibitor trametinib with a pan-EGFR inhibitor or FGFR inhibitor led to tumor shrinkage in epithelial-like and mesenchymal-like KRAS mutant patient-derived xenograft models, respectively [54]. A similar EGFR/ERBB3-mediated feedback reactivation of MAPK signaling was observed following MEK inhibition in colorectal cancer, consistent with its epithelial-like phenotype [55,56]. However, clinical trials combining EGFR inhibitors with MEK inhibitors have resulted in, at most, disease stabilization, with no objective responses due to toxicities [57].

Another approach to treating KRAS mutant cancers is to identify drugs specifically targeting KRAS-dependent or -independent tumors by exploiting their vulnerabilities. A drug-repurposing computational in silico screen inspecting a drug-gene signature network, with a reverse oncogenic-specific signature, identified decitabine, an analogue of nucleotide cytidine, as a candidate that can generate a KRAS G12V-reversing signature [58]. The vulnerability of KRAS-dependent pancreatic cancer cells to decitabine was based on the impairment of de novo pyrimidine biosynthesis. For KRAS-independent tumors, metabolic reprograming bypasses the dependency for mutant KRAS. As described above, the frequent co-mutations with KEAP1 or NRF2 observed in these KRAS-independent tumor cells lead to dependency on increased glutaminolysis for survival, with one study demonstrating that the pharmacological inhibition of glutaminase results in tumor shrinkage in vivo [33].

### 5.2. KRAS Dependency and Response to KRAS G12C Inhibitors

In recent years, significant progress has been made in developing direct inhibitors of mutant KRAS, particularly the G12C variant. These compounds covalently bind to cysteine 12 within the switch 2 pocket of the GDP-bound form of KRAS G12C [22,59,60,61]. Biochemically, these agents capture KRAS in its inactive GDP-bound state and lock it in this conformation, thereby inhibiting downstream pro-tumorigenic signaling that translates to preclinical antitumor responses. In clinical trials, two potent, selective, and irreversible small-molecule KRAS G12C inhibitors, sotorasib (AMG 510) and adagrasib (MRTX849), have demonstrated promising single-agent activity across cancers harboring KRAS G12C [7,8]. In spite of the initial positive response in patients with KRAS G12C mutant lung cancer treated with these inhibitors, both adaptive and acquired resistance can limit their efficacy. To date, multiple preclinical studies have reported various mechanisms of resistance to KRAS G12C inhibitors. These studies overall suggest the reactivation of MAPK signaling and/or mutant KRAS-independent activation of the PI3K pathway as common modalities to bypass KRAS G12C blockade, ultimately resulting in KRAS G12C inhibitor resistance [42,62,63,64].

Several mechanisms for the reactivation of MAPK signaling, as well as for mutant KRAS-independent activation of PI3K signaling, have been reported [42,60,62,63,64]. As presented earlier, the inhibition of the MAPK pathway induces feedback reactivation of MAPK signaling via multiple activated RTKs in various types of cancer cell lines. Some of these RTKs can also activate PI3K signaling. Additionally, genetic alterations of RTK-RAS-MAPK signaling genes have been found in patients with clinically acquired resistance to KRAS G12C inhibitors [65,66]. While the analysis of re-biopsied tissue samples or circulating tumor DNA (ctDNA) obtained at the time of acquiring resistance to MRTX849 largely identified secondary mutations in KRAS, multiple genetic alterations relating to the activation of PI3K and MAPK signaling were also observed. These include the amplification of the *KRAS* G12C allele, *MET* amplification, activating mutations in *NRAS*, *BRAF*, *MAP2K1*, and *RET*, oncogenic fusions involving *ALK*, *RET*, *BRAF*, *RAF1*, and *FGFR3*, and loss-of-function mutations in *NF1* and *PTEN* [65]. Furthermore, single-cell RNA sequencing analysis of KRAS G12C mutant NSCLC cell lines revealed that treatment with the KRAS G12C inhibitor ARS1620 induced gene expression and protein synthesis of KRAS G12C [63]. The newly synthesized mutant KRAS G12C proteins can be activated by EGFR or aurora kinase A (AURKA) signaling, leading to its drug-insensitive GTP-bound state, thereby acquiring KRAS G12C inhibitor resistance via the reactivation of MAPK signaling. Thus, the reactivation of MAPK signaling and mutant KRAS-independent activation of PI3K signaling are achieved by various means, conclusively resulting in mutant KRAS independency and resistance to KRAS G12C inhibitors. SHP2, a nonreceptor protein tyrosine phosphatase, is part of the signaling adaptor complex that mediates RAS activation downstream of RTKs. Therefore, the inhibition of SHP2 can shut down signaling from RTKs to wild-type RAS following MAPK inhibition, including in the setting of KRAS G12C inhibitor treatment [67,68,69,70]. Currently, the combination of SHP2 inhibitors with KRAS G12C inhibitors is under evaluation (NCT04185883, NCT04330664).

EMT is also associated with intrinsic and acquired resistance to KRAS G12C inhibitors in NSCLC [42,62]. Gene set enrichment analysis (GSEA) between KRAS G12C inhibitor-sensitive and intrinsically resistant cells identified a KRAS dependency signature that was significantly associated with sensitivity to AMG510 [42]. In line with this, KRAS G12C inhibitor sensitivity was associated with a higher expression of KRAS protein and KRAS GTP activity. Moreover, GSEA revealed an increased mesenchymal phenotype harbored by resistant cells. Similar to the MEK inhibitor case, the activated RTKs following KRAS G12C inhibition were mainly ERBB2/3 in epithelial-like KRAS G12C mutant and FGFR/AXL in mesenchymal-like KRAS mutant lung cancer cell lines. In cell lines with intrinsic or acquired resistance to AMG510, combined inhibition of KRAS G12C and SHP2 led to more durable ERK inhibition by abrogating RTK-SHP2-wild-type RAS-mediated MAPK pathway reactivation. However, the efficacy for the combination of AMG510 with the SHP2 inhibitor was impaired by SHP2-independent PI3K-AKT pathway activation mediated by the IGF1R pathway. These results suggest that the combination of KRAS G12C inhibitors with SHP2 inhibitors will not be satisfactory enough to treat patients with mesenchymal-like KRAS G12C mutant cancer (Figure 2).

In a phase II study of sotorasib, results of exploratory analyses of molecularly defined subgroups demonstrated that the highest overall response rate among patients with LKB1 mutation and wild-type KEAP1 was 50%. On the contrary, patients with KEAP1 mutant tumors appeared to derive less benefit from sotorasib, with a median progression-free survival of 5.5 months for those with wild-type LKB1 tumors and 2.6 months for those with KEAP1 and LKB1 double-mutant tumors. While it is conceivable that these clinical trial results are related to KRAS dependency and metabolic rewiring, the precise mechanisms remain to be elucidated.

## 6. Conclusions

Although KRAS mutations are an indispensable step in the pathogenesis of KRAS mutant cancers, tumor cells can eventually acquire KRAS independency by activating multiple signaling networks, as summarized in Figure 3. Biologically, the activation of bypass signaling is presumably beneficial for KRAS mutant cancer cells to survive by coping with hypoxia and accelerating metastasis. Translationally, the activation of bypass signaling has been shown to be a major cause of resistance to molecular targeted therapies such as EGFR and ALK [71]. Given that we are in the dawn of the era of druggable KRAS, an understanding of the mechanism of KRAS independency will be critical for patient selection and the development of optimal treatment strategies.

## Figures and Tables

**Figure 1 cancers-13-05081-f001:**
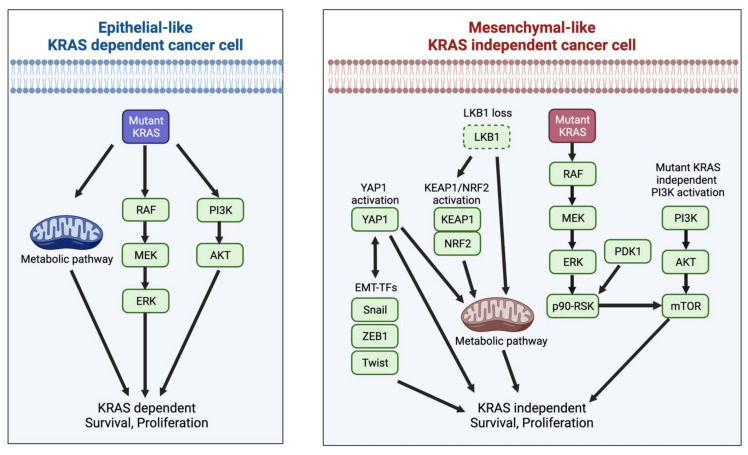
Heterogeneity of KRAS mutant cancer characterized by EMT and KRAS dependency. The role of mutant KRAS in proliferation and survival is variable in tumor cells driven by KRAS mutations. While KRAS-dependent cancer cells show an epithelial-like phenotype, KRAS is dispensable for the survival of mesenchymal-like KRAS mutant cancer cells. KRAS mutant cancer cells acquire mutant KRAS independency by various means: activation of RSK-mTOR pathway and/or KEAP1-NRF2 pathway, YAP activation, LKB1 loss, and mutant KRAS-independent PI3K activation. These bypass signaling pathways are involved in activation of survival signaling pathways, rewiring of metabolic pathways, and induction of EMT. EMT transcription factors also promote KRAS independency. The activated bypass signaling networks and EMT are interdependent. Abbreviations: EMT, Epithelial-to-mesenchymal transition; EMT-TFs, Epithelial-to-mesenchymal transition transcription factors; KEAP1, Kelch-like ECH-associated protein 1; KRAS, v-Ki-ras2 Kirsten rat sarcoma viral oncogene homolog; LKB1, liver kinase B1; mTOR, mammalian target of rapamycin; NRF2, Nuclear factor erythroid 2-related factor 2; PI3K, Phosphatidylinositol 3-Kinase; RSK, Ribosomal S6 kinase; YAP, Yes-associated Protein 1.

**Figure 2 cancers-13-05081-f002:**
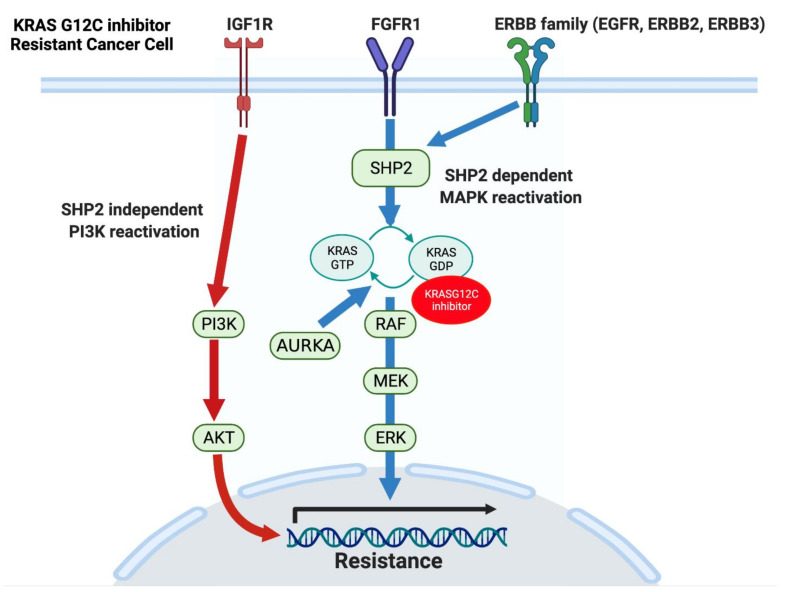
Mechanisms of resistance to KRAS G12C inhibitors. Inhibition of MAPK signaling by G12C inhibitors induces feedback reactivation of MAPK via multiple receptor tyrosine kinases, including FGFR1 and ERBB family members. While FGFR1 and ERBB3 reactivate the MAPK signaling pathway through adaptor protein SHP2, IGF1R leads to SHP2-independent activation of the PI3K signaling pathway. Additionally, activation of EGFR and Aurora kinase following KRAS G12C inhibitor treatment leads to protein synthesis and activation of KRAS G12C. The newly synthesized and activated G12C proteins are resistant to KRAS G12C inhibition due to their GTP-bound form. Furthermore, mutations in MAPK proteins can induce constitutive MAPK activation in the presence of KRAS G12C inhibitors. Activation of PI3K and MAPK leads to transcription of genes related to cell proliferation and survival (black arrow). Abbreviations: EGFR, Epidermal growth factor receptor; ERBB, Erb-B2 Receptor Tyrosine Kinase; FGFR1, Fibroblast growth factor receptor 1; IGF1R, Insulin-like growth factor 1 receptor; MAPK, Mitogen-activated protein kinase; SHP2, Src homology region 2 domain-containing phosphatase 2.

**Figure 3 cancers-13-05081-f003:**
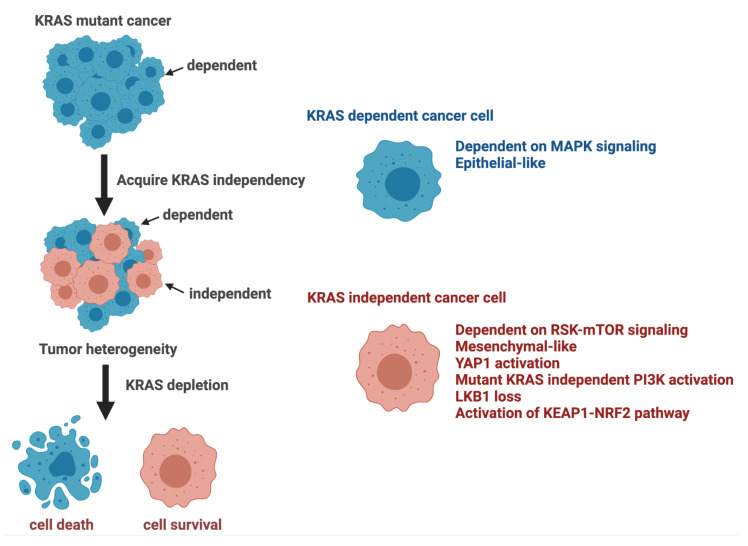
Phenotypes and activated pathways characteristically observed in KRAS-dependent and independent KRAS mutant cancer cells. Abbreviations: KEAP1, Kelch-like ECH-associated protein 1; KRAS, v-Ki-ras2 Kirsten rat sarcoma viral oncogene homolog; LKB1, liver kinase B1; mTOR, mammalian target of rapamycin; NRF2, Nuclear factor erythroid 2-related factor 2; PI3K, Phosphatidylinositol 3-Kinase; RSK, Ribosomal S6 kinase; YAP, Yes-associated Protein 1.

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
