# Peer review of "Escaping KRAS: Gaining Autonomy and Resistance to KRAS Inhibition in KRAS Mutant Cancers"

_cancers, 2021, doi:10.3390/cancers13205081_

Round 1
Reviewer 1 Report
This is a very clearly written review that is appropriately focussed on a topic that is of significant current interest.
A few parts of the manuscript could be improved to provide an outstanding contribution:
- Use of KRAS vs Kras is not always consistent. It may be that the former is used for human and the latter for mouse, but some instances seem more random.
- The sentence in the summary "Notably, mutant KRAS specific inhibitors appear to be effective only in KRAS dependent tumors" is not really clear. The G12C inhibitors do not even work against KRAS dependent tumors that are driven by other mutations.
- The sentence "Notably, tumor cell show negligible apoptosis by KRAS depletion in 2D culture, but become highly dependent on KRAS in 3D culture or in vivo (20)" is making an important point. Unfortunately the cited reference (#20) does not seem to support that.
- Figure 1 is not very clear. The legend states that the "activated bypass signaling networks and EMT are interdependent", but the figure as drawn really shows them as separated.
- In Figure 2, there is a bypass arrow for SHP2 dependent MAPK reactivation, but this arrow points to the nuclear membrane, which is not informative. For a more mechanistic description of the pathway, the authors should show whether the literature indicates that the reactivation of ERK occurs directly at the level of ERK, or via MEK, or by another route.
Author Response
POINT-BY-POINT REPLY TO Reviewer 1
- Use of KRAS vs Kras is not always consistent. It may be that the former is used for human and the latter for mouse, but some instances seem more random.
We have checked the consistency throughout the manuscript and corrected it.
- The sentence in the summary "Notably, mutant KRAS specific inhibitors appear to be effective only in KRAS dependent tumors" is not really clear. The G12C inhibitors do not even work against KRAS dependent tumors that are driven by other mutations.
We have edited the sentence to “Notably, KRAS G12C specific inhibitors appear to be effective only in tumors dependent on mutant KRAS for their survival.”
- The sentence "Notably, tumor cell show negligible apoptosis by KRAS depletion in 2D culture, but become highly dependent on KRAS in 3D culture or in vivo (20)" is making an important point. Unfortunately the cited reference (#20) does not seem to support that.
We apologize for the mistake and have updated the reference.
- Figure 1 is not very clear. The legend states that the "activated bypass signaling networks and EMT are interdependent", but the figure as drawn really shows them as separated.
We have edited Figure 1 to show the interdependency between YAP1 and EMT transcription factors.
- In Figure 2, there is a bypass arrow for SHP2 dependent MAPK reactivation, but this arrow points to the nuclear membrane, which is not informative. For a more mechanistic description of the pathway, the authors should show whether the literature indicates that the reactivation of ERK occurs directly at the level of ERK, or via MEK, or by another route.
The reactivation of MAPK signaling was mediated by RTKs (reference 69). We have clarified the point in Figure 2.

Reviewer 2 Report
KRAS is one of the most frequently mutated oncogenes in human cancers. The vast majority of KRAS alterations are single nucleotide point mutations. However, different cancers bearing KRAS mutations demonstrate different dependencies on this oncogene as some tumors are dependent on KRAS activity, where others are entirely independent. In this review, Adachi et al. describe the molecular mechanism by which cancer escape KRAS-dependency, including the role of different signaling and metabolic pathways that bypass KRAS. Additionally, they describe the role of the EMT program as a mechanism for KRAS resistance.
Comments:
In chapter 2, the authors describe the three different KRAS mutants: 12,13 and 61. However, they need to emphasize that the frequency of the mutations is different as the G12 mutation is by far more abundant.
In chapter 3, they describe that “knockdown or knockout of the oncogene induces robust apoptosis”. This statement needs to be referenced.
In chapter 4.2, the siRNA screen study and the dependency on other RAS need to be referenced.
At the end of chapter 4.3, “oncogenic KRAS promotes micropinocytosis to transport extracellular proteins as an amino acid source for the TCA cycle to sustain tumor growth [72]”. Check if this is the correct reference number.
In chapter 4.4, the transcription factors are designated as” EMT-transcription factors” (take out “related”).
Figure 1 is unclear; This cartoon demonstrates the signaling pathway that activates selective transcription factors; how does this show KRAS independence? What does each color represent?
More updated references should be provided, for example, reference 20.
Author Response
POINT-BY-POINT REPLY TO Reviewer 2
- In chapter 2, the authors describe the three different KRAS mutants: 12,13 and 61. However, they need to emphasize that the frequency of the mutations is different as the G12 mutation is by far more abundant.
We added a sentence “Among them, codon 12 is by far the most frequently observed.”
- In chapter 3, they describe that “knockdown or knockout of the oncogene induces robust apoptosis”. This statement needs to be referenced.
We have added references, presenting EGFR and BRAF knockdown induce apoptosis in EGFR mutant and BRAF V600E mutant cancer cell lines, respectively (New references 16 and 17).
- In chapter 4.2, the siRNA screen study and the dependency on other RAS need to be referenced.
We have added the reference, presenting RAS dependency identified by an siRNA screen (Reference 21).
- At the end of chapter 4.3, “oncogenic KRAS promotes micropinocytosis to transport extracellular proteins as an amino acid source for the TCA cycle to sustain tumor growth [72]”. Check if this is the correct reference number.
We apologize for the mistake and updated the reference (Reference 40). We also apologize for the error of “micropinocytosis,” as KRAS promotes macropinocytosis, not micropinocytosis. The error was corrected throughout the manuscript.
- In chapter 4.4, the transcription factors are designated as” EMT-transcription factors” (take out “related”).
We agree with your suggestion. The word “related” is taken out from the sentence in chapter 4.4 and throughout the manuscript where relevant to EMT.
- Figure 1 is unclear; This cartoon demonstrates the signaling pathway that activates selective transcription factors; how does this show KRAS independence? What does each color represent?
Thank you for the comment. Figure1 shows that multiple bypass pathways are activated in mesenchymal like KRAS independent cancer cells. To avoid confusion, we have unified the color. Also, signaling in KRAS dependent cancer cells was inserted to clarify the distinction between KRAS dependent and independent cells.
- More updated references should be provided, for example, reference 20.
We appreciate the comment and updated the references.
